# Concept for the Treatment of Class III Anomalies with a Skeletally Anchored Appliance Fabricated in the CAD/CAM Process—The MIRA Appliance

**DOI:** 10.3390/bioengineering10050616

**Published:** 2023-05-19

**Authors:** Lutz D. Hodecker, Reinald Kühle, Frederic Weichel, Christoph J. Roser, Christopher J. Lux, Carolien A. J. Bauer

**Affiliations:** 1Department of Orthodontics and Dentofacial Orthopedics, University of Heidelberg, Im Neuenheimer Feld 400, D-69120 Heidelberg, Germany; lutz.hodecker@med.uni-heidelberg.de (L.D.H.); christoph.roser@med.uni-heidelberg.de (C.J.R.); christopher.lux@med.uni-heidelberg.de (C.J.L.); 2Department of Oral and Maxillofacial Surgery, University of Heidelberg, Im Neuenheimer Feld 400, D-69120 Heidelberg, Germany; reinald.kuehle@med.uni-heidelberg.de (R.K.); frederic.weichel@med.uni-heidelberg.de (F.W.)

**Keywords:** angle Class III, orthodontic anchorage procedure, computer aided design, case report, retrognathia, growth and development, orthodontic appliances

## Abstract

Objective: Intermaxillary elastics, anchored skeletally, represent a promising concept for treatment in adolescent patients with skeletal Class III anomalies. A challenge in existing concepts is the survival rate of the miniscrews in the mandible or the invasiveness of the bone anchors. A novel concept, the mandibular interradicular anchor (MIRA) appliance, for improving skeletal anchorage in the mandible, will be presented and discussed. Clinical case: In a ten-year-old female patient with a moderate skeletal Class III, the novel MIRA concept, combined with maxillary protraction, was applied. This involved the use of a CAD/CAM-fabricated indirect skeletal anchorage appliance in the mandible, with interradicularly placed miniscrews distal to each canine (MIRA appliance), and a hybrid hyrax in the maxilla with paramedian placed miniscrews. The modified alt-RAMEC protocol involved an intermittent weekly activation for five weeks. Class III elastics were worn for a period of seven months. This was followed by alignment with a multi-bracket appliance. Discussion: The cephalometric analysis before and after therapy shows an improvement of the Wits value (+3.8 mm), SNA (+5°), and ANB (+3°). Dentally, a transversal postdevelopment in the maxilla (+4 mm) and a labial tip of the maxillary (+3.4°) and mandibular anterior teeth (+4.7°) with gap formation is observed. Conclusion: The MIRA appliance represents a less invasive and esthetic alternative to the existing concepts, especially with two miniscrews in the mandible per side. In addition, MIRA can be selected for complex orthodontic tasks, such as molar uprighting and mesialization.

## 1. Introduction

In the orthopedic treatment spectrum of orthodontics, the therapy of skeletal Class III anomalies represents a challenge in clinical routine due to the long-lasting skeletal growth, the strongly pronounced hereditary component, and the functional correlations [1]. A skeletal Class III is a type of jaw/facial anomaly in which the lower jaw is positioned too far forward in relation to the upper jaw, or the upper jaw is positioned too far back. A combination of both is also possible. Skeletal Class III can lead to a variety of problems, including difficulty chewing and speaking, uneven tooth attrition, periodontal disease, and aesthetic impairment of the facial profile with accompanying psychosocial handicaps.

Skeletal Class III can result from genetic factors, growth disorders, or injury [2]. Continuous and in-depth diagnostics, an appropriate treatment plan with appliances, and the need to control growth beyond the terminal stage of growth are the key elements of skeletal Class III therapy [2,3].

In addition to basic orthodontic procedures and adaptation of a myofunctional balance, orthopedically acting forces play a central role in orthodontic correction of Class III anomalies [4]. The fundamental orthopedic mechanisms include growth inhibition or growth regulation of the mandible and stimulation of maxillary growth in terms of maxillary protraction and/or transverse postdevelopment [5]. The challenge of applying orthopedic forces in a targeted manner is the selection and consistency of an appropriate approach to forces. The current literature distinguishes between extraoral and intraoral approaches. The intraoral anchorages are additionally subdivided into dental, skeletal, and hybrid forms. Purely extraoral appliances include head-chin caps and, in combination with intraoral anchorage, face masks [4,6]. Intraoral anchorage options include dental-anchored Class III mechanisms, such as removable appliances or Class III elastics on the multiband. Different approaches to skeletally anchored maxillary protraction are described in the literature [7,8]. A distinction is made between bone-anchored maxillary protraction, in which four bone anchors are used [9], Class III bone-anchored elastics, in which two bone anchors or two miniscrews are used in the mandible and a hyrax in the maxilla [10], skeletally anchored facial masks, which use two bone anchors or two miniscrews in the maxilla and a face mask, and the bone-anchored rapid maxillary expansion, which combines a hybrid hyrax in the maxilla with a face mask, a Mentoplate in the mandible, or an indirect skeletal mandibular anchorage [11,12,13]. The presented MIRA appliance belongs to the category of indirect skeletal anchorages.

The selection of the appropriate appliance depends on the dental age, the stage of development, the severity and shape of the Class III anomaly, and the patient’s cooperation. While early treatment of Class III is often necessary in the first mixed dentition phase, this is sometimes not sufficient in the case of strong mandibular growth and makes follow-up treatment necessary in the second mixed dentition phase. If skeletal Class III is not detected until the second mixed dentition phase, therapy should be initiated directly [3]. In the context of regular treatment starting from the second mixed dentition phase, skeletally anchored Class III appliances appear to be statistically advantageous over purely dentally supported appliances because a light continuous force is applied directly to the jawbone while the dentition is bypassed, resulting in higher skeletal changes [7,8,9,10,11]. However, these skeletally anchored Class III mechanics are characterized by an invasive surgical procedure. General anesthesia is usually required for insertion and removal of the anchorage plates. Insertion should only be performed by an experienced surgeon, as the anatomy of the bone must be recontoured with the plate to ensure the best possible fit. Common risks include inflammation, swelling, pain around the plates, and loosening of the plates. In contrast, placement of mini-implants requires only local anesthesia. Removal can even be done without local anesthesia. Various regions are generally suitable for inserting mini-implants in the mandible. These include the buccal shelf, the retromolar space and the interradicular area. While the first two regions often have good bone availability for screw placement [14,15], their posterior location makes them rather unsuitable as indirect skeletal anchorage for receiving Class III elastics. Interradicular bone availability is highly dependent on position, root location, and anatomy [16]. In particular, the insertion site and the small distance between the miniscrew and the root are considered risk factors for screw loss. According to a recent meta-analysis by Tepedino et al. 2020, the ideal interradicular insertion site in terms of bone availability is between the first and second molar (M1-M2) and first and second premolar, 5 mm below the enamel-cement interface [17]. Beyond this, predrilling is recommended. However, loss rates in the anterior region (canine—first premolar) are lower than in the posterior region (M1–M2) [16]. 

This case report highlights a modern indirect skeletal anchorage appliance for the mandible that was fabricated using the computer aided design and computer aided manufacturing (CAD/CAM) process (3Shape Appliance Designer™, 3Shape A/S, Copenhagen, Denmark). The MIRA appliance (mandibular interradicular anchor, Ortholize GmbH, Nienhagen, Germany) is intended to represent another option for indirect skeletal anchorage, which is particularly characterized by its wearing comfort, easy handling, and simple care.

## 2. Diagnosis

The 10-year-old patient, together with her mother, presented for consultation at the Department of Orthodontics at Heidelberg University Hospital on the advice of her general dentist. The patient was under regular dental supervision. There were no abnormalities in the patient’s general medical history. Her height growth was above the 97th percentile and her physical development was well ahead of her age group. Radiographic findings were normal. All permanent teeth—except the wisdom teeth—had already erupted. There was no hereditary clustering of Class III.

The clinical findings were dentitio praecox and a Class III anomaly with a narrow apical base in the maxilla and a caudal tongue rest position. In addition, there was a lateral crossbite at 2.4 to 2.6 with a mesial occlusion and a mandibular shift to the left. The overjet and overbite were 0 mm due to a frontal crossbite at 1.2 and 2.2. The skeletal deviation was associated with a labial tipping of the maxillary anterior teeth (+11.3°). Evaluation of the cephalometric lateral radiograph showed a mesial basal relation (Wits: −7.2 mm) and a neutral to horizontal growth pattern. Use of the cervical vertebral maturation (CVM) method for the assessment of mandibular growth provided staging between 3 and 4. The patient is therefore around the growth peak [18] (Figure 1). 

## 3. Treatment

### 3.1. Treatment Objectives

The patient was informed of the findings and treatment options were evaluated in a chairside decision process. A combined skeletal Class III therapy consisting of a hybrid hyrax in the maxilla and the novel skeletally anchored MIRA appliance in the mandible was considered the best treatment option. Skeletal anchorage of the hybrid hyrax was achieved via two paramedian-placed miniscrews. Dentally, the appliance was supported at 1.6 and 2.6. Additional hooks were placed on the molar shells to hold Class III elastics. In the mandible, interradicular miniscrews were placed between the premolars for skeletal anchorage (3.4–3.5 and 4.4–4.5). The MIRA appliances were also dentally anchored to the premolars and had a hook on either side at the level of the marginal gingiva of the canines to receive Class III elastics. Activation of the hybrid hyrax followed a modified Alt-RAMEC protocol for five weeks, with twice-daily screwing. The interval for the use of intermaxillary Class III elastics was set at seven months. The aim was to inhibit mandibular growth with simultaneous maxillary growth promotion in anterior–posterior and transverse planes without dental side effects.

### 3.2. Treatment Alternatives 

There are several approaches available for the treatment of an adolescent with a skeletal Class III. A distinction is made between therapy before the growth peak, around the growth peak, and after the growth peak. Appliances to choose from include functional orthodontic appliances, facemasks, head-chin caps, Class III elastics, and skeletally anchored Class III elastics. In severe cases, surgical correction of the jaw position is also an option, but this is usually performed only when it can be assumed that the patient is no longer growing. To estimate the skeletal age, the CVM method can provide information [18]. 

Before the growth peak, functional orthodontic appliances, such as the Fränkel type III, are often used. In severe cases, maxillary protraction with maxillary anchorage and extraoral facemask may also be necessary. The functional orthodontic approach is the least invasive. Successful treatment of Fränkel type III depends on the patient’s cooperation. Since the patient was already advanced in growth, the Fränkel type III was considered too weak to compensate for the transversal discrepancy in the maxilla. 

Maxillary protraction by means of the face mask could probably have achieved a good treatment result. A tooth-supported appliance in the maxilla could have resolved the transversal discrepancy. However, this would have side effects on the posterior teeth. A skeletal hybrid hyrax could also have been used in combination with a facemask. This could have reduced the side effects on the posterior teeth. However, as the patient refused to wear the face mask for psychosocial reasons, these treatment options were discarded.

The concepts were discussed and evaluated with the patient and mother while taking advantages and disadvantages into consideration. Since the patient was well advanced in growth for her age, the permanent teeth (especially 3.5, 3.4, 4.5 and 4.4) had already erupted, and she was already around peak growth, a decision was made in favor of the innovative MIRA approach. Constant force application of weak forces through intermaxillary Class III elastics was considered most effective for this narrow time window. Furthermore, the parents and the patient were informed about a possible orthognatic surgery in case of unfavorable growth (Figure 2).

### 3.3. Treatment Progress

The treatment schedule was six to twelve months and included overcorrection of the overjet. The treatment was started with the insertion of miniscrews. Two screws were inserted paramedian in the maxilla and interradicular between 3.4 and 3.5 and between 4.4 and 4.5 in the mandible. One week after insertion, digital impressions were taken using intraoral scans. The scan provided the basis for the appliance design, which was designed using a computer-aided 3D planning technique. In the maxilla, a hybrid hyrax was designed with connection to the miniscrews and shells at 1.6 and 2.6. Additional hooks were attached to the shells for intermaxillary elastics. In the mandible, two MIRA appliances were designed for indirect anchorage with the miniscrews. They had hooks extending mesially, which were later used to receive intermaxillary elastics. The appliances were fabricated using a computer-assisted laser melting process. For this case series, the appliances were digitally designed and manufactured by OrthoLIZE GmbH (Nienhagen, Germany). The insertion of the appliances was performed after previous cleaning and drying with a mineral cement (3M™ Ketac™ Cem radiopaque, 3M Company, Minnesota, USA) and screwing in the plate screws for fixation to the miniscrews. Conditioning of the tooth surfaces was not necessary. Class III elastics with 3/16 inch medium (1.3 N) were inserted, which the patient was asked to change three times a day and wear for 22 h a day. At the same time as the elastics, a modified alt-RAMEC protocol was performed in the maxilla for five weeks, in which the patient turned the transverse expansion screw open one turn twice daily for one week and closed one turn twice daily for one week. In the fifth week, transverse expansion was performed by twisting it open one turn twice daily to the desired width. The intermaxillary elastics were worn for seven months. This was followed by the removal of the appliances and the changeover to a multibracket appliance for alignment and closure of the remaining gaps (Figure 3).

### 3.4. Treatment Results

After five weeks of modified alt-RAMEC protocol in the maxilla and seven months of intermaxillary Class III elastics, there was a positive overjet of +5 mm. The cephalometric lateral radiograph, taken for control, showed an increase in the SNA angle of 5° and in the ANB angle of 3° compared to the beginning. The Wits value improved by 3.8 mm. The mandibular angle decreased by 1.7°. A slight increase in the proclined position of the upper anterior teeth was also observed (Figure 4). The patient coped very well with the appliances. She was pleased with the invisibility and ease of handling. Periodontal hygiene was maintained throughout the treatment.

## 4. Discussion

Based on the presented case report, a novel CAD/CAM-manufactured appliance for the treatment of a skeletal Class III anomaly should be presented and evaluated in terms of clinical workflow and efficacy. The presented MIRA appliance follows the principle of skeletally anchored maxillary protraction and mandibular growth inhibition according to De Clerck and is expected to be less invasive and more suitable for practical use [7,9,19,20]. The lower invasiveness is achieved by placing miniscrews in the maxilla in the T-zone and in the mandible interradicularly in the premolar region under local anesthesia. The higher surgical effort required for the concepts described in the literature is thereby eliminated [9,12,13].

The MIRA appliance demonstrated skeletal Class III improvement in an adolescent patient who was around the growth peak at the time of treatment. The duration of treatment was seven months. The cephalometric evaluation before and after treatment showed an improvement in basal relation, with a change in Wits value of +3.8 mm, and ANB of +3°. In addition, there was a change in the anterior tooth axes. Proclination of the anterior teeth was observed in the maxilla and mandible, which was also clinically manifested in gap formation. Due to the low restriction of orofacial esthetics and the simplicity of the application of Class III elastics, the patient showed a high degree of adherence. Furthermore, no complications regarding miniscrew stability, MIRA appliance retention, or periodontal inflammation were noted during this period.

The results of skeletally anchored Class III elastics correspond to the study situation and can be rated as successful [7,8,9,10,11,12,13,21,22,23,24,25,26,27,28,29,30,31]. Advantages seem to be the lower invasiveness compared to bone anchorage plates, the low restriction of orofacial esthetics compared to facemasks, and the high degree of customizability of the appliance [19]. Furthermore, it has been shown that this form of skeletal anchorage can also be used for complex difficult orthodontic tasks, such as orthodontic gap closure in the mandible or the alignment of displaced teeth (Figure 5). In these cases, a configuration of the MIRA with one screw and the enclosure of at least two teeth seems to be sufficient. This means that in addition to the orthopedic therapy task, an orthodontic intervention can also be combined with the MIRA. 

The dependence on the dental stage of development could prove to be disadvantageous. The MIRA appliance can only be used after complete eruption of the lower canine and at least one lower premolar. In general, Class III therapy should be started as early as possible to have a high chance of success [3,4,5]. It is recommended to initiate a functional orthodontic Class III therapy before the MIRA appliance, especially during the first mixed dentition phase when the MIRA appliance cannot yet be used. If strong Class III growth persists during the second mixed dentition phase, a switch to the MIRA appliance should be made shortly before, around, or at the latest shortly after the growth peak. Furthermore, the anatomical conditions in the mandible are challenging when placing the miniscrews. In addition to the lower bone volume and interradicular insertion, there is often a lack of sufficient width of the attached gingiva at this age. The risk of root damage or screw loosening is increased with low interradicular bone availability [16,17]. In anatomically difficult situations, where only one screw can be placed per quadrant, the risk of dental side effects is to be expected, as the MIRA appliance in this configuration does not provide sufficient protection against rotation. The authors therefore recommend a configuration with two screws per quadrant, if possible, contrary to the concept presented by Gera et al. and as in the case report presented [11].

The authors point out that only a limited selection of suitable screws for the MIRA appliance are currently available. The screws should be small in diameter due to the interradicular insertion but should allow a coupling via an internal connection thread [15]. The screw head should be as gracile as possible to allow insertion of the appliance. Thus, if the insertion is too horizontal, there is a risk that the rigid appliance will not fit over the teeth and the screw. Further studies must follow to evaluate the most favorable positioning of the screw and the screw design (Figure 6).

## 5. Conclusions

The MIRA appliance represents an alternative to the previous orthopedically acting skeletally anchored Class III appliances, especially with two mini-screws in the mandible per side. Due to the high degree of individualization, the minimally invasive procedure and the low restriction of orofacial esthetics, the MIRA appliance can be integrated into everyday practice as a supplement to conventional orthodontic Class III appliances. In addition, the MIRA appliance can be selected for complex orthodontic tasks, such as molar uprighting and mesialization. However, this case report is only a case report, and these promising results still need to be evaluated in detail in a clinical study, especially regarding which configuration proves to be most advantageous.

## Figures and Tables

**Figure 1 bioengineering-10-00616-f001:**
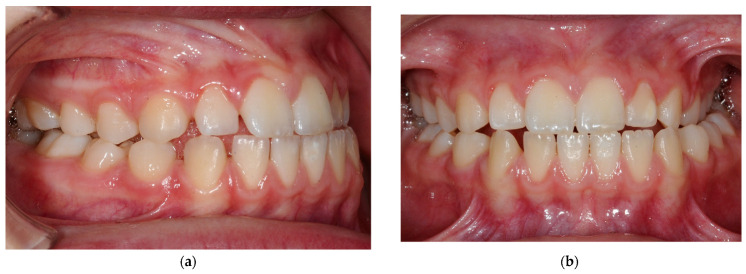
Initial diagnosis: (**a**) intraoral view of the right side in occlusion, (**b**) intraoral view from the front in occlusion, (**c**) lateral cephalometric image, (**d**) cephalometric analysis before treatment, (**e**) lateral profile before treatment, (**f**) en face with smile before treatment.

**Figure 2 bioengineering-10-00616-f002:**
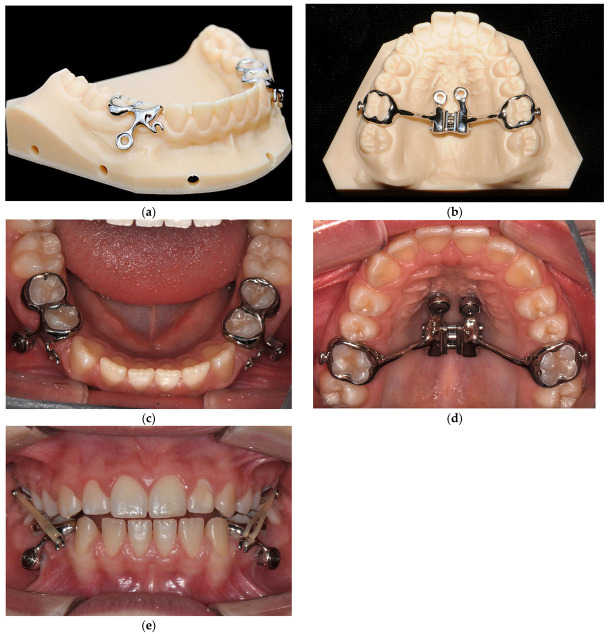
(**a**) MIRA appliance on lower model before insertion, (**b**) hybrid hyrax on upper model before insertion, (**c**) intraoral image of the lower jaw with inserted MIRA appliance on both sides, (**d**) intraoral image of the upper jaw with inserted hybrid hyrax, (**e**) intraoral frontal image with inserted Class III elastics.

**Figure 3 bioengineering-10-00616-f003:**
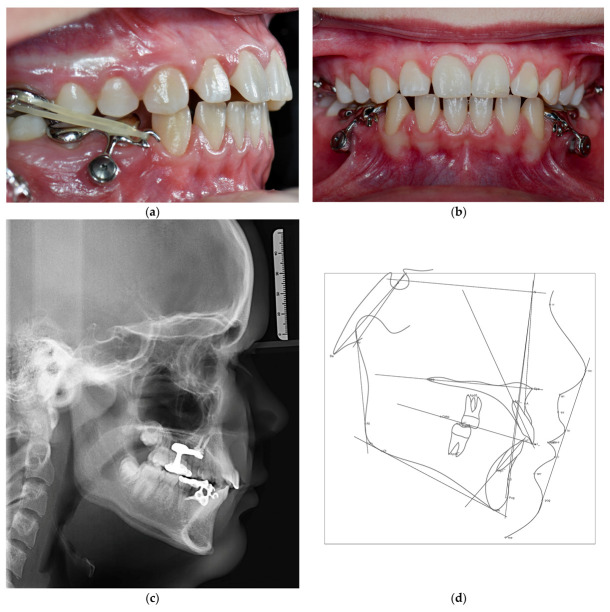
Situation after a five week modified alt-RAMEC protocol and a seven month maxillary protraction by using Class III elastics; (**a**) intraoral image of the right side with sagittal overcorrection of 4 mm; (**b**) intraoral image frontal with transversal overcorrection of 3 mm; (**c**) cephalometric lateral image, (**d**) cephalometric analysis after treatment, (**e**) lateral profile after treatment, (**f**) En face with smile after treatment.

**Figure 4 bioengineering-10-00616-f004:**
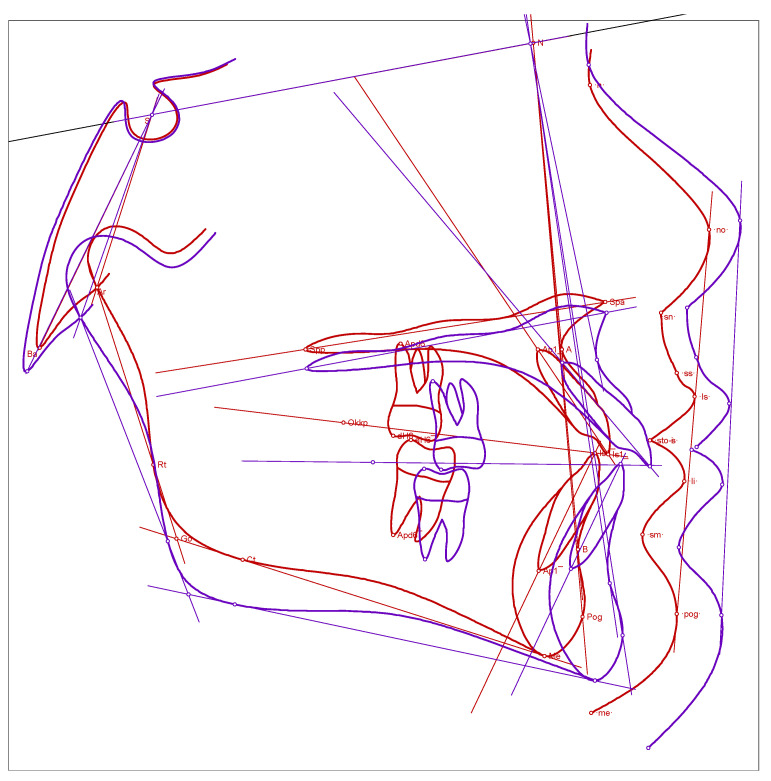
Overlay of the cephalometric images. The red drawing corresponds to the initial situation, the purple drawing to the condition after seven months of treatment. The overlay plane was the Sella—Nasion line.

**Figure 5 bioengineering-10-00616-f005:**
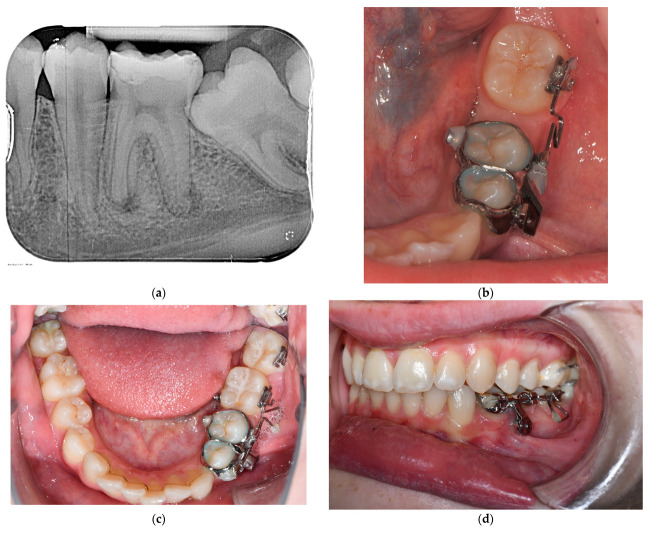
Exemplary presentation for orthodontic tasks (mesialization of lower molars, straightening and alignment of displaced teeth) the MIRA appliance. (**a**) Dental X-ray in region 3.5, 3.6, 37: apical inflammation and extensive filling 3.6, retention 3.7 with posterior crowding and close positional relationship to 3.8, (**b**) condition after extraction of 3.6 and insertion of a MIRA appliance with a delta loop for mesialization of 3.7, (**c**) top view of the lower jaw: condition after complete mesialization and uprighting of 3.7 and 3.8, (**d**) lateral view: condition after complete mesialization and uprighting of 3.7 and 3.8, (**e**) radiological check after mesialization and uprighting of 3.7 and 3.8, (**f**) top view of mandible: final finding.

**Figure 6 bioengineering-10-00616-f006:**
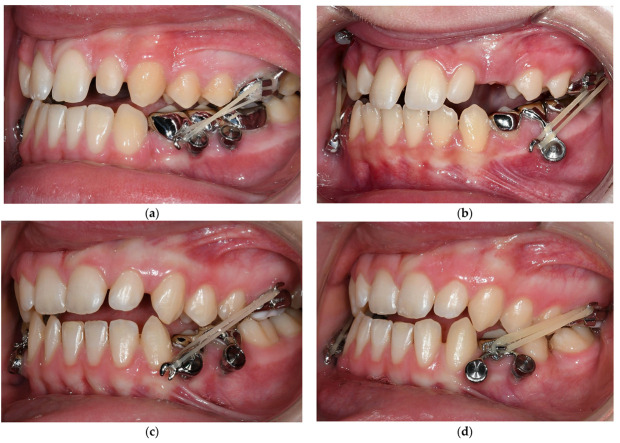
Modifications of the MIRA appliance; (**a**) two interradicular pins and dental support on three teeth; (**b**) one interradicular pin and dental support on three teeth; (**c**) two interradicular pins with support on two teeth; (**d**) two interradicular pins with support on one tooth.

## Data Availability

The data presented in this study are available on request from the corresponding author.

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
