# Peer review of "Concept for the Treatment of Class III Anomalies with a Skeletally Anchored Appliance Fabricated in the CAD/CAM Process—The MIRA Appliance"

_bioengineering, 2023, doi:10.3390/bioengineering10050616_

Round 1
Reviewer 1 Report
Dear authors,
Thank you for the possibility to review this interesting article. A few points need to be clarified prior to the publication of this manuscript.
Keywords: Please check your keywords to ensure that they are MeSH terms.
Introduction:
The introduction is fairly weak on background and justification, and how this study contributes to the literature.
Spell out all acronyms the first time that they are used (e.g.: CAD/CAM)
Material & Methods
No specific details regarding patient consent. Your manuscript does not contain a complete IRB statement regarding ethics board approval. Original articles need to contain a statement about the Helsinki Declaration of 1975, as in the example given here: “This study was approved by the human subjects ethics board of XXXXX and was conducted in accordance with the Helsinki Declaration of 1975, as revised in 2013.
Discussion:
The Discussion is slightly shallow and it is difficult to determine what points are based on the author's study, their conjecture, or previously published literature. The Limitations section should be expanded to include concerns raised in Weaknesses.
It is good!
Author Response
Dear Reviewer,
Thank you for your effort and time to prepare a review for our study. We took your comments and suggestions for improvement very seriously and worked through them point by point.
Keywords: Please check your keywords to ensure that they are MeSH terms.
- The keywords were checked and corrected.
Introduction:
The introduction is fairly weak on background and justification, and how this study contributes to the literature.
- The introduction was revised with background information and studies. In addition, the present case report was classified in the current literature.
Spell out all acronyms the first time that they are used (e.g.: CAD/CAM)
- The acronyms were written out when they were first used.
Material & Methods
No specific details regarding patient consent. Your manuscript does not contain a complete IRB statement regarding ethics board approval. Original articles need to contain a statement about the Helsinki Declaration of 1975, as in the example given here: “This study was approved by the human subjects ethics board of XXXXX and was conducted in accordance with the Helsinki Declaration of 1975, as revised in 2013.
- No statement regarding the ethics board was obtained in advance because the patient case did not primarily involve research. However, since this patient case was retrospectively well treated in our opinion, we would like to present it in the context of a case report. Consent for publication was obtained from the patients prior to writing. Thus, from our point of view, the Declaration of Helsinki was observed.
Discussion:
The Discussion is slightly shallow and it is difficult to determine what points are based on the author's study, their conjecture, or previously published literature. The Limitations section should be expanded to include concerns raised in Weaknesses.
- The discussion was revised to highlight our points more clearly. In addition, the limitation area has been expanded.We already mentioned that there are different approaches to anchor intermaxillary class III elastics with the aim of a low side-effect and non-invasive thrapy. We present an approach where an interradicular skeletal anchorage is to be strengthened via a CAD/CAM fabricated framework. However, this case report is only a case report and these promising results still need to be evaluated in detail in a clinical study. The limitation or dependence of the dental age has to be discussed. It can only be done after the canine and the 1st premolar have erupted. Pretreatment with functional appliances is necessary. However, if this is not sufficient, the MIRA appliance can be a reasonable alternative. Disadvantages on the dentition must be excluded. Since there is no protection against rotation on the screw, with continuous weak force application, it is best to work with 2 implants per side or with a wide support on the teeth. Which configuration proves to be most advantageous must be found out in a clinical study.
Thank you for the great suggestions for improvement and the comprehensive review, we have tried to implement all suggestions.
With kind regards
the authors

Reviewer 2 Report
Dear Authors,
in attach you find some remarks. I hope they could be useful for your work.
Best regards

Author Response
Dear Reviewer,
Thank you for your effort and time to prepare a review for our study. We took your comments and suggestions for improvement very seriously and worked through them point by point.
Please add in the title a link to the ‘case report’ you presented.
- The words were supplemented
Abstract. The paragraph ‘materials and methods’ do not match with the case report presented. Please rename in ‘Description’ or ‘Clinical case’ or similar. The paragraph ‘results’ should be renamed in ‘Discussion’ or similar. In the paragraph ‘Conclusion’ please avoid hazardous suggestions about the clinical use of MIRA. They are not justified by this single case report.
- You are absolutely right; we have included in the discussion and as an outlook the respective points noted.
Paragraph ‘Diagnosis’. In lines 67-74 you cited briefly the height growth and physical development. Did you perform any evaluation of sheletal age and growth? Cervical evaluation or wrist? It could help readers to understand the correct time of intervention with this technique.
- We added the Cervical Vertebral Maturation (CVM) Method.
Lines 101-105. When you suggest to use MIRA? Timing of intervention should be discussed. Treatment of Class III defects depends even on residual growth of the patient…? Please discuss.
- We suggest MIRA shortly before / during and shortly after the growth peak to specifically slow down mandibular growth. We have included this point.
Figure 3. Radiographs (Fig 1 and 3) are not comparable. The vertical line is not respected. Please attach the cephalometric analysis before and after the treatment, even superimposed to the radiographs. As you know, in this case it is important to evaluate the profile of the patient. Please add photos of lateral profile and smile before and after treatment.
- We have added the images and evaluations you requested.
Thank you for the great suggestions for improvement and the comprehensive review, we have tried to implement all suggestions.
With kind regards
the authors
Round 2
Reviewer 1 Report
All comments have been satisfactorily resolved.
It is satisfying.